# Stimulatory Effects of (+)-Epicatechin on Short- and Long-Term Memory in Aged Rats: Underlying Mechanisms

**DOI:** 10.3390/nu17223611

**Published:** 2025-11-19

**Authors:** Israel Ramirez-Sanchez, Veronica Salas-Gutierrez, Rosa Ordoñez-Razo, Pilar Ortiz-Vilchis, Claudia Calzada-Mendoza, Veronica Najera, Patricia Mendoza-Lorenzo, Guillermo Ceballos, Francisco Villarreal

**Affiliations:** 1Seccion de Estudios de Posgrado e Investigacion, Escuela Superior de Medicina, IPN, Mexico City 11340, Mexico; psicologaverosalas@gmail.com (V.S.-G.); raliporvi@hotmail.com (P.O.-V.); cccalzadam@yahoo.com.mx (C.C.-M.); gceballosr@ipn.mx (G.C.); 2Unidad de Investigacion Medica en Genetica Humana, Hospital de Pediatria, Centro Médico Nacional Siglo XXI, Mexico City 06720, Mexico; romaorr@yahoo.com.mx; 3School of Medicine, UCSD, La Jolla, CA 92093, USA; vinajera@ucsd.edu (V.N.); fvillarr@ucsd.edu (F.V.); 4Division Academica de Ciencias Basicas, Universidad Juarez Autonoma de Tabasco, Villahermosa 86660, Mexico; patricia.mendoza@ujat.mx; 5VA San Diego Health Care, San Diego, CA 92093, USA

**Keywords:** flavanol, epicatechin, memory, learning

## Abstract

**Background/Objectives**: We previously reported that the flavanol (+)-epicatechin (+Epi) enhances adult mice short-term working memory and neurogenesis. This study aimed to characterize the effects of +Epi on short- and long-term memory, to modulate mitochondria structure/function, oxidative stress (OS) and inflammation associated cytokines in the hippocampus and pre-frontal cortex of aged rats. **Methods**: Experiments were conducted using aged (23 month old) male Sprague Dawley rats. The control group (n = 6/group) were exposed to vehicle (water) only while the treated group, was provided +Epi at 1 mg/kg/day by oral gavage for 8 weeks. Open-field recognition tests were used to evaluate short- and long-term memory. The hippocampus and frontal cortex were sampled and citrate synthase activity, ATP levels, mitochondrial proteins, cytokines (IL-1β, IL-6, TNF-a and IL-11), protein carbonylation, lipid peroxidation (malonaldehyde; MDA), superoxide dismutase 2 (SOD2), glutathione peroxidase (GPx) and catalase activity were quantified. **Results**: There was a significant improvement in both short- and long-term memory in the +Epi treated group vs. controls. Mitochondrial bioenergetics also improved with treatment as determined by increased citrate synthase activity and ATP content. Relative levels of the mitochondrial proteins mitofilin and complex V increased with +Epi. +Epi suppressed protein carbonyls and MDA levels. OS buffering systems were significantly enhanced with +Epi as per increases in SOD2, GPx and catalase enzyme activities. +Epi also decreased pro-inflammatory and stimulated anti-inflammatory cytokines vs. controls. **Conclusions**: Results demonstrate +Epi improves mitochondrial function, reduces OS and inflammation in the hippocampus and cortex leading to improved short- and long-term memory in aged animals providing evidence for possible mechanisms of action.

## 1. Introduction

Aging is a natural process that over time leads to deleterious effects on neural integrity and cognition. A complex interplay between biochemical and physiological factors, social influences and life choices can contribute to cognitive decline with age [1]. Deficits in conceptual reasoning, processing speed, executive functioning, short-term and long-term memory typically appear with aging. These functional changes are underpinned with multiple structural alterations in the brain [2]. Atrophy is accompanied by detrimental changes such as amyloid deposition [3], dopamine receptor depletion [4], medial temporal lobe and hippocampus deterioration, thinning of the pre-frontal cortex and reduced functional connectivity between cortices develop with aging [5,6,7] and are thought to directly contribute to a decline in cognitive capacity. However, the underlying mechanisms driving structural alterations remain undetermined.

The brain has a high energy demand and through mitochondria based oxidative phosphorylation, ATP is generated. As ATP is produced, reactive oxygen species (ROS) are also generated which lead to oxidative stress (OS). As the brain ages, mitochondrial dysfunction ensues producing high levels of free radicals and thus, OS [8]. Unmitigated OS has been recognized as a deleterious systemic component of the aging process, implicated in the beginning and progression of neurodegeneration and cognitive decline. ROS can be efficiently degraded by buffering systems. Components of the buffering systems include the enzymes superoxide dismutase (SOD), catalase, and glutathione peroxidase (GPx) which metabolize ROS [9]. However, with aging, these buffering systems become less efficient and thus, an imbalance between the rates of production and ROS removal leads to OS induced cell damage, neurodegeneration and cognitive impairment [10,11]. With elevated OS the brain microglia and astrocytes release high levels of pro-inflammatory cytokines such as IL-1β, IL-6, and TNFα while reducing those with anti-inflammatory potential such as IL-10 [12,13]. Ultimately, OS driven chronic inflammation contributes neurodegeneration and age-related brain vulnerability and cognitive impairment. In the context of aging tissues and organs, this process is also known as inflammaging.

Multiple studies have provided evidence for the possible protective role of flavonoids (i.e., quercetin, and epicatechin) to counteract aging and/or disease associated loss of brain structure/function [14,15]. Flavonoid rich dietary interventions have reported to yield benefits on cognition via antioxidant, anti-inflammatory and direct neuroprotective actions [16]. The flavanols (-)-epicatechin (-Epi) and its enantiomer +Epi are both found in cacao. While -Epi is readily abundant, +Epi is found in much lesser concentrations [17]. However, both enantiomers have been found to upregulate the expression of neuronal developmental and differentiation factors such as NeuN, DCX, NGF, and MAP2, as well as neurofilament NF200, increasing markers of neurogenesis and improving short-term memory in adult mice [18]. Interestingly, +Epi appeared as more effective vs. -Epi in stimulating activators of neurogenesis as per its stronger stimulation of nitric oxide (NO) synthesis through eNOS phosphorylation.

This study was designed to investigate the impact of +Epi treatment on short- and long-term memory while also assessing OS markers (protein carbonyls and malondialdehyde levels), pro-inflammatory cytokines and mitochondrial function in pre-frontal cortex and hippocampus of normal aged male rats as per their potential mechanistic role in driving age associated cognitive deficits.

## 2. Materials and Methods

### 2.1. Study Design and Animal Model

Approval was obtained from UCSD’s Animal Care and Use Committee (IACUC) and all methods were carried out in compliance with relevant institutional, Federal and ARRIVE guidelines and regulations. The animal model of aging was approved by the funding agencies and UCSD and was implemented using 3 month old, male Sprague Dawley rats purchased from Charles River Inc. that aged in house at a temperature of 20–25 °C on a 12-h light/dark cycle to 23 months while being provided regular chow and water at libitum. At 23 months of age, rats were randomly allocated in 2 groups: (1) control animals only given vehicle (water by gavage) (n = 6) and, (2) +Epi (1 mg/kg/day in water by gavage) treated animals (n = 6) for a period of 8 weeks. To determine the number of animals used in the present study, we based ourselves on the results reported in our recent publication [17], where we analyzed the effects induced by epicatechin on front paw grip strength where an effect size of 1.88 with power equal to 0.834 was found when compared to a group of control animals yielding an n = 6 (number of animals used in the present study). We have previously reported on the effectiveness of 1 mg/kg/day of +Epi to activate neurological responses and thus, selected this dose to use [18]. +Epi was obtained as a gift from Epirium Bio Inc. (San Diego, CA, USA), (previously known as Cardero Therapeutics, Inc.). A detailed description on the synthesis, chemical characterization and purity of +Epi can be found in the article by Moreno-Ulloa et al., [19]. After completion of the treatment period, animals were subject to cognitive testing (see below), then euthanized, brain samples collected, the cortex and hippocampal regions were dissected as per our previous reported methods [20] and stored at −80 °C for biochemical analysis. All tissue processing and biochemical assays were performed in a blind manner and in a randomized order. Replicate samples were averaged before statistical analysis.

### 2.2. Locomotor Activity and Open Field Task (OFT)

To prevent behavior alterations due to exposure to a new environment and to familiarize animals with the environment, they were transported to the testing room during 3 consecutive days and left in this location for 1 h each day prior to formal testing [21]. Locomotor activity and anxiety-like behavioral responses to a novel environment were measured in an open field apparatus using the ANY-maze software 6300-0800-2019 (SD Instruments, Inc). The apparatus consisted of an acrylic 50 cm (length) × 50 cm (width) wall surrounded by a 50 cm high acrylic wall. Two areas were virtually defined in the apparatus using the ANY-maze software, the periphery (outer zone 10 cm from the wall) and the central area (the rest of the OF) (Figure 1A). Animals were gently placed in the center of the area at the beginning of the test and allowed to freely move in the quadrants of the area to explore the environment for 10 min period. Animal movements and paths were recorded using the ANY-maze video tracking system. The video recordings were used to measure total distance traveled and the time spent in each area [22].

### 2.3. Object Recognition Task (ORT)

Animals were exposed to the testing room as in the OFT to avoid behavior alterations triggered by exposure to a new environment. Animals were also individually habituated to the open empty area for exploration for 5 min consecutively on two days. The object sets used were of the same size, different color and shape (cubes, pyramids and cylinders) plastic objects which were located at the same distance from the wall arena. We also counterbalanced the objects by shifting positions and side assignments. Animals were familiarized for 5 min with 2 identically aligned objects (A + A) on the following day which were placed in the open field at 10 cm from the walls. Rats were then sent back to their cages for 1 h and reintroduced after 1 object was exchanged (A + B) and allowed to explore for 5 min to evaluate short-term memory. After 24 h, object B was exchanged for C and animals allowed to explore for 5 min (long-term memory) (Figure 1D). Pointing the nose toward the object at a distance of <1 cm and/or touching it with the nose was considered as exploration behavior. Sitting close or turning around to the object was not considered as exploration [18]. Two different individuals blind to the experimental conditions quantified video recordings. Time spent between the novel (TN) and the time sum of both objects novel and familiar (TF) [(TN)/(TN + TF)] defined object recognition.

### 2.4. Protein Carbonyl Measurements

Carbonylation was used as a surrogate indicator of tissue protein OS. Brain samples (~10 mg) were homogenized in 250 μL of cold buffer (50 mM 4-morpholineethanesulfonic acid, pH 6.7, with 1 mM EDTA) and centrifuged (10,000× *g*) for 15 min at 4 °C. Supernatants of homogenates were gathered and incubated for 15 min at room temperature with streptomycin sulfate (1% of final concentration). Samples were centrifuged for 10 min at 4 °C (6000× *g*). Using a colorimetric protein carbonyl assay kit (Cayman Chemicals, 10005020) carbonylation levels were measured in supernatants following kit instructions using a spectrophotometer at 360 nm. Samples were tested in duplicate.

### 2.5. Measurement of Malondialdehyde for Lipid Peroxidation

Malondialdehyde (MDA) quantitation was used to measure lipid peroxidation. Brain samples (~10 mg) were washed in cold PBS, homogenized on ice in 150 μL of lysis solution (RIPA buffer, ThermoScientific, 89901), supplemented with protease inhibitors and centrifuged for 10 min at 4 °C at 1600× *g*. Using a colorimetric assay kit (TBARS) supernatants were collected and tested in duplicate at room temperature to measure malondialdehyde bound to thiobarbituric acid (TBA) according to kit instructions (Cayman Chemicals, 10009055) by spectrophotometry at 540 nm.

### 2.6. SOD2 Activity

Tissue samples (~10 mg) were rinsed with cold PBS to remove blood cells. Tissues were homogenized in 150 μL of sucrose buffer (0.25 M sucrose, 10 mM Tris, 1 mM EDTA, pH 7.4). The homogenates were sonicated on an ice bath for 5 min and centrifuged at 10,000× *g* for 60 min at 4 °C. The supernatants were collected, supplemented with KCN (at final concentration of 1 mM) as to inactivate non-SOD2 activities and prepared to measure enzyme activity using a colorimetric kit according to manufacturer instructions (Dojindo Molecular Tech., WST). Absorbance was read at 450 nm in a spectrophotometer. All samples were tested in duplicates and measured at room temperature.

### 2.7. Catalase Activity

Tissue samples (~10 mg) were rinsed with cold PBS to remove blood cells, then homogenized in 150 μL of cold buffer (50 mM potassium phosphate, pH 7.4 containing 1 mM EDTA) and centrifuged at 10,000× *g* 15 min at 4 °C. The supernatant was used to measure catalase activity using a colorimetric assay detection kit (Cayman Chemicals, 707002) according to the manufacturer’s instructions. Absorbance was read at 540 nm in a μQuant spectrophotometer. All samples were tested in duplicates and measured at room temperature.

### 2.8. Glutathione Peroxidase Activity

Tissue samples (~10 mg) were rinsed with cold PBS to remove blood cells. Samples were homogenized in 150 μL of cold buffer (50 mM Tris-HCl pH 7.5, containing 5 mM EDTA and 1 mM DTT) and centrifuged at 10,000× *g* for 15 min at 4 °C. The supernatants were collected to measure GPx activity using a colorimetric detection assay kit (Cayman Chemicals, 703102) according to the manufacturer’s instructions. Absorbance was read at 340 nm using a μQuant spectrophotometer. All samples were tested in duplicates and measured at room temperature.

### 2.9. Citrate Synthase Activity

Brain samples (~25 mg) were homogenized in cold extraction buffer (250 μL, 20 mM Tris-HCl, 140 mM NaCl, 2 mM EDTA, and 0.1% sodium dodecyl sulfate) supplemented with protease inhibitors (Sigma-Aldrich, P2714), 5 mM Na_3_VO_4_ and 3 mM NaF. Lysates were centrifuged for 15 min at 4 °C at 10,000× *g*. Supernatants were obtained and used to measure enzymatic activity as the rate of production of the mercaptide ion based on conversion of acetyl-coenzyme A (CoA) and oxaloacetate into CoA-SH. CoA-SH in the presence of 5,5-disthiobis-2-nitrobenzoic acid produces mercaptide ion. Using a μQuant spectrophotometer at 412 nm duplicate samples were analyzed at room temperature.

### 2.10. ATP Assay

The levels of ATP were determined using an ATP kit (Life Technologies, A22066). About 10 mg of tissue was homogenized in cold isolation buffer (200 μL, 5 mM HEPES pH 7.2, 225 mM mannitol, 75 mM sucrose, 1 mM EGTA, with protease inhibitors). Samples were centrifuged for 5 min at 4 °C at 1500× *g* and supernatants were collected to measure ATP using a luminometer (in duplicate) at room temperature following kit instructions.

### 2.11. Cytokines Measurements

Tissue samples (~10 mg) were rinsed with cold PBS to remove blood cells. Then, tissues were homogenized in 150 μL of cold buffer (50 mM potassium phosphate, 1 mM EDTA, pH 7.4), supplemented with protease inhibitors, and centrifuged at 13,000× *g* for 15 min at 4 °C. The supernatant was collected and used for measurements of total protein content (Bradford method) and for the measurements of IL-1β, IL-6, and TNF-α, by Luminex xMAP Milliplex magnetic-bead immunoassay (MilliporeSigma by design), and quantified in a Bioplex plate reader (Bio-Rad), while IL-11 was measured using the colorimetric ELISA assay kit (Novus Biologicals, NBP3-06795) and detected measuring absorbance at 450 nm using a μQuant spectrophotometer. Cytokines to be evaluated were based on their reported [23] pro- and anti-inflammatory actions in brain. However, cytokines are known to exert varying “opposing” actions on the basis of the model used [23]. The panel tested ultimately provides a picture albeit limited, to the cytokine responses to treatment. Samples were evaluated in duplicates at room temperature. Concentrations of cytokines were calculated using the standard curve for each kit.

### 2.12. Western Blotting

Hippocampus and frontal cortex tissue samples (~25 mg) were homogenized in 250 μL of lysis buffer (1% Triton X-100, 200 mM Tris, 140 mM NaCl, 2 mM EDTA, and 0.1% sodium dodecyl sulfate) with protease and phosphatase inhibitor cocktails (P2714; Sigma-Aldrich), supplemented with 0.15 mM PMSF, 5 mM Na_3_VO_4_ and 3 mM NaF, as previously described [20]. Homogenates were sonicated in an ice bath tank sonicator (Fisherbrand, USA) for 15 min and centrifuged (10,000× *g*) for 10 min at 4 °C. The total protein content was measured in the supernatant using the Bradford method. A total of 40 μg of protein were loaded onto a 4–15% Mini-PROTEAN^®^ TGX™ Precast Protein Gel (Bio-Rad), electrotransferred to a PVDF membrane using a Semi-Dry system (16 V, 90 min) (Bio-Rad). Membranes were incubated for 1 h in blocking solution (5% nonfat dry milk in Tris-buffered saline plus 0.1% Tween 20), followed by overnight incubation at 4 °C with primary antibodies diluted at 1:1000 or 3 h at room temperature. Antibodies used included complex V (Invitrogen, 43-9800), mitofilin (Abcam, ab110329) or GAPDH (Cell Signaling, 2118). Membranes were washed (3 × 5 min) in Tris-buffered saline plus 0.1% Tween 20 and incubated for 1 h at room temperature with specific horseradish peroxidase (HRP)-conjugated secondary antibodies diluted 1:10,000 in blocking solution. Immunoblots were developed using an enhanced chemiluminescence (ECL) detection kit (Amersham-GE). Densitometric analyses of bands were performed using ImageJ 2.0 software.

### 2.13. Statistical Analysis

All results are presented as mean ± standard error of the mean (SEM) and analyzed by unpaired *t*-test to determine differences between each group means. *p* values < 0.05 were considered statistically significant.

## 3. Results

### 3.1. OFT and ORT

To account for test induced altered behavioral activity total ambulatory distance was measured (Figure 1B). Results from the total distance traveled demonstrate no differences in locomotor activity between control and +Epi treated animals. The OFT, which is used as an indicator of anxiety-like behavior, demonstrated a significant reduction in outer vs. center zone times with +Epi treatment vs. controls (Figure 1C). ORT (Figure 1D) assessed short- and long-term memory in aged rats. Rats explored two identical objects for the total exploration time during the training phase. One and 24 h after training, short-term and long-term memory sessions were conducted using one familiar and one novel object. No significant differences in total exploration time were observed between groups across training session (Figure 1E), indicating comparable exploratory activity and motivation during testing. In short- and long-term memory tests (Figure 1F,G), the +Epi group demonstrated increased time or exploration, demonstrating improved recognition memory. +Epi also significantly improved the object preference score vs. controls in both short (Figure 1I) or long-term memory (Figure 1J) measures.

### 3.2. Protein and Lipid Oxidation

Carbonyl proteins were significantly decreased by ~55% with +Epi treatment in hippocampus (Figure 2A; 6.3 ± 0.7 nmol per mg of protein) vs. controls (11 ± 0.8 nmol per mg of protein) and by ∼43% (Figure 2B) in cortex (4.8 ± 0.6 nmol per mg of protein) vs. controls (8.8 ± 0.6 nmol per mg of protein). MDA levels were significantly reduced by ~68% in hippocampus (Figure 2C) with +Epi treatment (3.0 ± 0.5 nmol per mg of protein) vs. controls (4.3 ± 0.4 nmol per mg of protein) and in cortex (Figure 2D) ∼66% (2.4 ± 0.4 nmol per mg of protein) vs. controls (3.6 ± 0.3 nmol per mg of protein).

### 3.3. SOD2, Catalase, and GPx Activities

As shown in Figure 3, catalase activity was significantly increased with +Epi treatment by ∼64% (2.4 ± 0.4 nmol per mg of protein) vs. controls (1.5 ± 0.3 nmol per mg of protein) in the hippocampus (Figure 3A) and by ~70% (2.0 ± 0.3 nmol per mg of protein) vs. (1.18 ± 0.3 nmol per mg of protein) in cortex (Figure 3B). SOD2 activity was significantly increased in animals with +Epi by ∼66% (2.0 ± 0.5 nmol per mg of protein) vs. controls (1.2 ± 0.3 nmol per mg of protein) in hippocampus (Figure 3C) and ∼90% (2.4 ± 0.8 nmol per mg of protein) vs. (1.25 ± 0.7 nmol per mg of protein) in cortex (Figure 3D). GPx activity was significantly increased with +Epi treatment by ∼56% (10.2 ± 0.7 nmol per mg of protein) vs. controls (6.5 ± 0.4 nmol per mg of protein) in hippocampus (Figure 3E) and ∼81% (9.23 ± 1.1 nmol per mg of protein) vs. (5.1 ± 0.3 nmol per mg of protein) in cortex (Figure 3F).

### 3.4. Mitochondrial Function

Mitochondrial function (Figure 4) was evaluated via citrate synthase activity (CSA) and ATP content determinations. CSA levels were significantly increased in hippocampus of animals with +Epi by ∼85% (2.12 ± 0.4 mmol per mg of protein) vs. controls (1.12 ± 0.2 mmol per mg of protein) (Figure 4A) and ∼91% (1.74 ± 0.07 mmol per mg of protein) vs. (0.88 ± 0.09 mmol per mg of protein) in cortex (Figure 4B). ATP content was significantly increased in the hippocampus of animals with +Epi by ∼51% (44.8 ± 6 nmol per mg of protein) vs. controls (29.5 ± 5 nmol per mg of protein) (Figure 4C) and ∼55% (36.9 ± 4 nmol per mg of protein) vs. (23.78 ± 4 nmol per mg of protein) in cortex (Figure 4D).

Using Western blots, we also evaluated the relative protein levels of mitofilin, a structural protein of mitochondrial cristae and complex V as an essential unit of the mitochondrial electron transport chain. Relative levels of both proteins increased in the hippocampus of animals with +Epi by ∼65% (mitofilin) and ~40% (complex V) vs. controls (Figure 4E). In frontal cortex (Figure 4F) both, mitofilin and CV were increased ∼30% with +Epi vs. controls.

### 3.5. Cytokines

We also evaluated the impact of +Epi treatment on the pro-inflammatory (IL-1β, IL-6 and TNF-a) and anti-inflammatory (IL-11) cytokines in aged rats. In hippocampus, +Epi treatment significantly reduced the levels of IL-1β (Figure 5A) by ~35% (6.28 ± 0.8 pg per mg of total protein) vs. controls (9.56 ± 1.1 pg per mg of total protein), IL-6 (Figure 5B), by ~33% (10.21 ± 1.7 pg per mg of total protein) vs. (15.25 ± 2.7 pg per mg of total protein) and TNFα (Figure 5C) by ~27% (6.88 ± 1.3 pg per mg of total protein) vs. (9.41 ± 1.1 pg per mg of total protein), while IL-11 (Figure 5D) increased by ~54% (256 ± 32 pg per mg of total protein) vs. controls (166 ± 24 pg per mg of total protein).

Similar results were obtained for frontal cortex (Figure 5E–H). +Epi treatment significantly reduced IL-1β (Figure 5E) ~35% (7.63 ± 0.5 pg per mg of total protein) vs. controls (11.73 ± 0.7 pg per mg of total protein), IL-6 (Figure 5F) by ~40% (13 ± 0.7 pg per mg of total protein) vs. (21.7 ± 0.8 pg per mg of total protein) and TNFα (Figure 5G) by ~25% (8.88 ± 0.7 pg per mg of total protein) vs. (11.81 ± 0.8 pg per mg of total protein), while IL-11 (Figure 5H) increased by ~56% (292.2 ± 51 pg per mg of total protein) vs. controls (187 ± 37 pg per mg of total protein).

## 4. Discussion

Unique findings from this study demonstrate that 8 weeks of oral supplementation with +Epi in aged rats is capable of improving cognitive function as per reduced anxiety-like behavior and improved short and long-term memory. Underpinning these positive effects are improvements in OS levels and related buffering systems, mitochondrial structure and function and cytokine profiles in hippocampus and frontal cortex.

As per the increasing aging profile of the general population, cognitive deficits have emerged as a leading challenge to modern society’s wellbeing. In recognition of such challenges and with the aim of reducing its burden, a major increase in research and therapeutic efforts targeting cognitive deficits has been committed by public and private entities. As per the limited success attained so far, a large amount of resources have been allocated to care for the afflicted populations [24]. Thus, the urgent need to find safe and effective therapies to reverse these trends. An important determinant of cognitive health and decline is derived from genetic determinants that may be difficult to modify. However, a greater level of recognition has been realized in identifying habits and practices that can be used to limit age associated cognitive decline. It has become apparent that factors such as dietary habits, physical activity, active social circles, limited exposure to polluted environments are all elements that can influence cognitive health and decline [25]. Even when such factors are modified for the better, there still remains a need to identify complementary strategies to sustain cognitive health.

So far, only a limited number drugs have been approved with the aim of mitigating cognitive decline [26,27]. Unfortunately, effects are at best, modest, several of these drugs carry significant adverse effects, can be costly and little is known about the sustained/long-term use adverse effects. Ideally, the identification of safe and effective compounds which can be taken over indefinite periods of time to prevent and/or reverse cognitive decline would be ideal.

It is readily apparent that specific components of human diets that are derived from natural sources can be very effective in promoting an overall healthier profile [28]. Current evidence suggests that components of a Mediterranean diet such as fish oils or foods rich in flavonoids can help individuals achieve this goal or reverse adverse tendencies that are instigated by unhealthy eating habits. Flavonoids are a group of natural compounds that are characterized by their polyphenolic structure and in general by the presence of color as in the case of berries. Flavonoids can be found in a variety of foods from fruits, tea leaves, grape skin, etc. [29]. There are four general subgroups of flavonoids. The flavanol subgroup is found in foods such as grapes, tea, berries, apples and cacao (chocolate) [29]. Diets that are balanced and include such elements in moderate amounts are recognized as health promoting [28].

Evidence indicates that the healthy effects of flavonoids can be realized in a variety of organs including the brain [29]. In humans, the consumption of flavonoid-containing foods has been investigated in relation to neurocognitive functioning with positive outcomes [30,31]. Consumption of specific flavonoid subtypes, such as flavanols (in cacao) and anthocyanins (in berries), have been associated with improved memory and executive function. Also, their combination of has been demonstrated to enhance executive function in healthy individuals and to confer neurocognitive protection in older adults [32]. Flavonoids from green tea [33], berries [34], grapes [35] and cacao [36] can induce neural plasticity and enhance memory and brain function. Flavonoids can also exert beneficial effects on cognitive function and neuroplasticity in early adulthood suggesting that their prolonged consumption may be neuroprotective in the long-term [31,37]. Flavonoids may also protect against age-related cognitive decline and neurodegenerative diseases such as Alzheimer’s and Parkinson’s by promoting the survival and growth of neurons, improving synaptic plasticity, and reducing neuroinflammation [38]. Overall, the regular consumption of flavonoid-rich foods is linked to better cognitive performance, slower brain aging, and a reduced risk of dementia [38].

As per the most recent evidence, cacao flavanols appear to promote cognitive health. The regular consumption of cocoa enriched with flavanol has been demonstrated to reduce age-related cognitive dysfunction including memory attention and executive function in normal elderly people or those with mild cognitive impairment [39,40,41]. Reports derived from the testing of young individuals yield modest increases in processing speed and attention [37]. Improvements in mood and mental fatigue are also reported with cocoa consumption [42]. Increased cerebral blood flow after cocoa intake are reported in fMRI studies [43]. Long-term consumption of cocoa-based diets have shown enhanced hippocampal neuroplasticity and neurogenesis in adult rodents, which is associated with enhanced cognition and emotional regulation [44]. Thus, there is strong supportive evidence as to the potential of cocoa flavanols to enhance brain functions.

Epi the most abundant flavanol in cacao and is found in two stereoisomeric forms (-) and (+) [45]. The beneficial effects of cocoa on cardiovascular health have been ascribed to the direct actions of Epi. Both forms of Epi are readily absorbed into the circulation and readily cross the blood-brain barrier [46]. Epi is able to increase brain blood flow and oxygenation [47]. Epi can be neuroprotective during acute hypoxia preserving cognition, mood [48], and short-term memory [49]. Epi has also been demonstrated to facilitate spatial memory processes and promote neuronal plasticity in the hippocampus and cortex [18,33]. In mice, Epi also increases brain angiogenesis and neuronal spine density, upregulates mRNA levels of proteins associated with learning in hippocampus [49]. Epi enhances the retention of spatial memory [18,50], while other flavonoids also appear to exert similar effects; for instance, (−)-epigallocatechin-3-gallate improves spatial learning and memory deficiencies in mouse models of aging and Alzheimer’s [51].

It is widely acknowledged that the aged brain may be particularly susceptible to oxidative damage due to the high levels of ROS production [52]. In addition, the age-associated decay in ROS neutralization capacity and prolonged exposure to OS in the brain can trigger inflammation and lead to the development of progressive memory impairment [53]. We previously reported that -Epi restores indicators of mitochondrial biogenesis, structure and activity in the pre-frontal cortex of aged (26 month-old) mice while also increasing OS buffering systems [54]. Using the same mouse model of aging, we also reported on the capacity of 4 weeks of Epi treatment (1 mg/kg/day) to significantly mitigate hippocampus OS, inflammation markers such as cytokines, hyperphosphorylation of tau protein, soluble β-amyloid protein levels while improving multiple cell survival linked endpoints, memory, anxiety-like behavior levels and systemic inflammation. These actions are in line with our previous reported effects of Epi where it can improve function in mitochondria through a variety of mechanisms such as limiting organelle swelling, restoring damaged cristae and stimulating biogenesis [55,56,57]. As the health of mitochondria improves, this reduces ROS generation and OS triggered inflammation. We also compared the effects of -Epi and +Epi on mouse frontal cortex-dependent short-term working memory and modulators of neurogenesis [18]. Male mice (3 month old) were provided water, (-)-Epi, at 1 mg/kg or +Epi at 0.1 mg/kg of body weight for 15 days. Results evidenced the stimulatory capacity of -Epi and +Epi on markers of neuronal proliferation and capillary density. Effects correlated positively with nitrate/nitrite stimulation by (-)-Epi and +Epi and enhanced eNOS phosphorylation. nNOS phosphorylation only increased with +Epi (18%). Neurofilament staining increased as well as NF200 and frontal cortex-dependent short-term spatial working memory. Results implied that both enantiomers, but more effectively +Epi, upregulate neurogenesis markers likely through stimulation of capillary formation and NO triggering, improvements in memory. While it can be argued that the effects of +Epi on neurological endpoints are not clearly superior to those previously reported with (-)-Epi [18], there is supportive in vitro evidence which suggests that +Epi (vs. (-)-Epi) has greater potency on mitochondrial endpoints [58] and if long-term (years) supplementation to humans was to be provided, ultimate effects on reducing age associated decline may prove to be superior. Long term clinical studies would need to be implemented to test for this possible scenario.

## 5. Conclusions

In summary, results from this study demonstrate the capacity of +Epi in the aged rats to positively impact cognitive functions as reflected by the enhanced both, short and long-term memory. Results hint at possible underlying mechanisms as per +Epi’s positive impact on hippocampus and frontal cortex OS levels and recovery of the ROS buffering system. Moreover, results demonstrate improved mitochondrial function which is associated with an increase in the expression of mitochondrial complex V levels and enhanced mitochondrial cristae protein mitofilin. As per published reports on the evaluation of +Epi effects in humans, these results support the exploration of the effects of +Epi in human trials [19].

## Figures and Tables

**Figure 1 nutrients-17-03611-f001:**
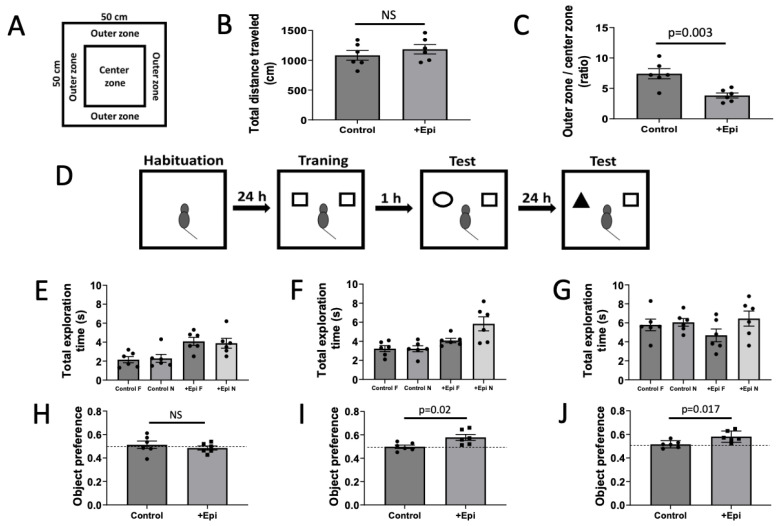
Positive effects of +epicatechin (+Epi) in treated aged rats on open field, total exploration time and object recognition tests. Panel (**A**) is the representation of an open field test. Panel (**B**) is the total distance traveled in the open field and (**C**), the time in outer vs. center zone ratio. Panel (**D**) is the image representation of the object recognition test. The schematic includes empty arena for habituation, 2 identical objects, cubes, cylinder and cube for short-term memory and pyramid and cube for long-term memory. Panel (**E**) reports on the total exploration time measured during the training, (**F**) on short-term memory, and (**G**) on long-term memory sessions of the object recognition test. Panel (**H**) is the graphic representation of the object preference test and scores obtained during training, short-term and long-term memory tests (**H**–**J**) respectively). Values reported compare chance score for object preference as = 0.5, (n = 6/group, mean ± SEM, *p* values are noted in each figure panel).

**Figure 2 nutrients-17-03611-f002:**
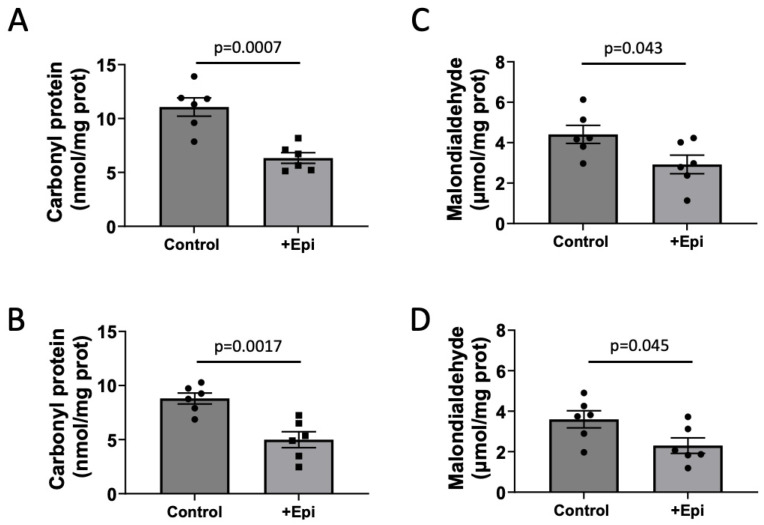
+Epicatechin (+Epi) induced reductions in protein (carbonylation) and lipid oxidation levels (malondialdehyde; MDA) in hippocampus and frontal cortex of treated aged rats. Protein carbonylation levels are reported for hippocampus (**A**) and frontal cortex (**B**) and changes observed in MDA in hippocampus (**C**) and frontal cortex (**D**), (n = 6/group, mean ± SEM, *p* values are noted in each figure panel).

**Figure 3 nutrients-17-03611-f003:**
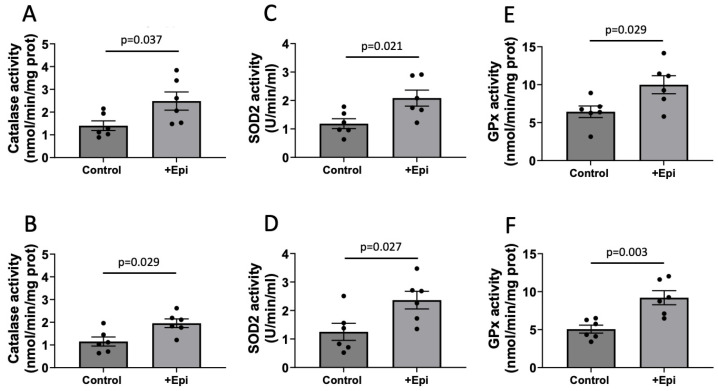
+Epicatechin induced stimulation of catalase, superoxide dismutase 2 (SOD2) and glutathione peroxidase (GPx) activity levels as evaluated in hippocampus and frontal cortex of treated aged rats. Catalase activity is reported for hippocampus (**A**) and frontal cortex (**B**) SOD2 enzyme activity in (**C**,**D**) while GPx activity are denoted in (**E**,**F**) respectively, (n = 6/group, mean ± SEM, *p* values are noted in each figure panel).

**Figure 4 nutrients-17-03611-f004:**
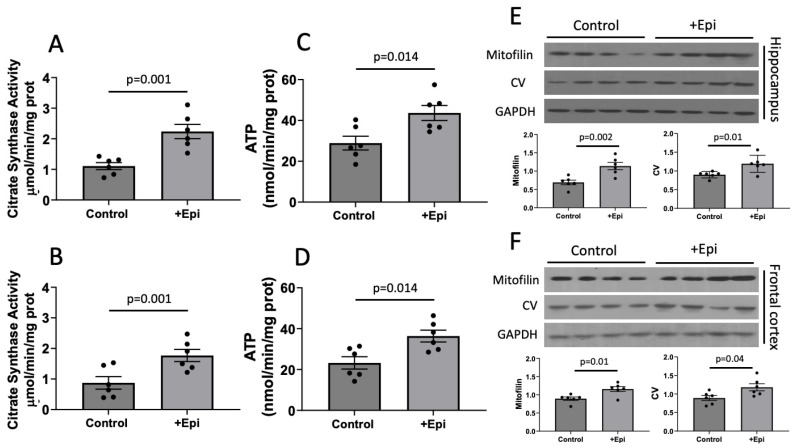
+Epicatechin (+Epi) induced stimulation of mitochondrial function (citrate synthase; CSA), ATP and protein levels in the hippocampus and cortex of aged rats. CSA levels are reported for hippocampus (**A**) and frontal cortex (**B**). Changes in ATP content are illustrated in (**C**,**D**) respectively. Representative Western blot images and plotted values for mitofilin and CV are denoted for hippocampus (**E**) and frontal cortex (**F**). relative protein levels were normalized using GAPDH values, (n = 6/group, mean ± SEM, *p* values are noted in each figure panel).

**Figure 5 nutrients-17-03611-f005:**
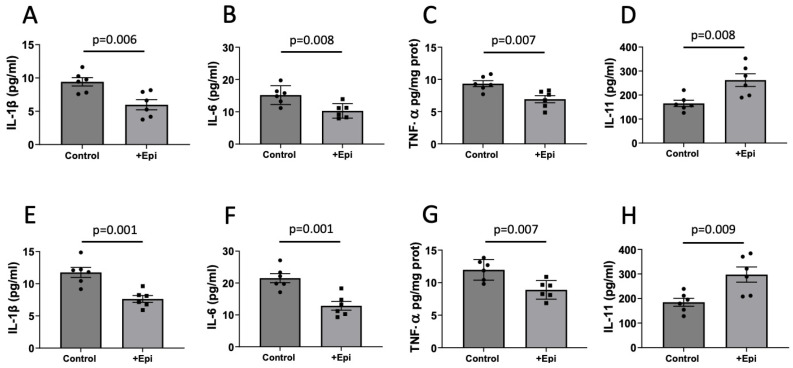
Positive effects of +epicatechin (+Epi) on pro- and anti-inflammatory cytokine levels in treated aged rats. Differences are reported for pro-inflammatory IL-1β (**A**), IL-6 (**B**), and TNF-α (**C**) and anti-inflammatory IL-11 (**D**) cytokines in hippocampus and frontal cortex, ((**E**–**H**) respectively) (n = 6/group, mean ± SEM, *p* values are noted in each figure panel).

## Data Availability

The raw data supporting the conclusions of this article are included including original Western blot images.

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
