# Peer review of "Stimulatory Effects of (+)-Epicatechin on Short- and Long-Term Memory in Aged Rats: Underlying Mechanisms"

_nutrients, 2025, doi:10.3390/nu17223611_

Round 1

Reviewer 1 Report

Comments and Suggestions for Authors

The current study investigates the impact of chronic intervention with pure (+)- epicatechin on aged rats, with a detailed investigation of hippocampus and frontal cortical modulation of mitochondria function, oxidative stress and inflammation. The study is novel and well executed, there are a few concerns and suggestions to improve the manuscript.

Comments:

  • The authors should specify what food sources contain (+)-epicatechin. Cocoa for example contain mainly (-) epicatechin ? What is the relative amount? It would be good to add a bit more on the rationale for looking at the (+) enantiomer. 
  • Rationale underlying the use of Object Recognition Tests to assess short-and long -term memory. Wouldn’t a task such as Radial Maze or Water Maze more appropriate for memory?
  • Add to the methods the rationale for a 1 mg/kg/day dose.
  • The sample size is rather small, would the authors be able to provide in the statistical section, power calculations used to estimate the sample size for the current study.
  • The initial part of the discussion, from line 310 to 367 seems a bit out of place, reads like an introduction. I suggest that the authors address first the results from their study and relationship with previous animal literature and then human literature (from line 368 onwards). Some of the sections in lines 310 to 367, might be used later on in the discussion, but need to be better integrated into the flow of the discussion of results.
  • In the discussion, a more in depth discussion on how (+)epicatechin might be able to affect mitochondrial function in the brain would be of value. There is also little effort to link how changes in inflammation and oxidative stress might link to mitochondrial function. The discussion requires more integration of results.. Is the flavanol (+) epicatechin affecting all of these pathways independently or is there a common mechanism?

Reviewer 2 Report

Comments and Suggestions for Authors

Israel Ramirez-Sanchez and colleagues investigated the effect of oral administration of (+)-epicatechin (1 mg/kg/day) on memory in elderly rats (23 months). After 8 weeks of administration, the authors reported an improvement in object recognition, and therefore in memory, without altering locomotion. In addition, the authors noted increased citrate synthase and ATP activity, an increase in myofilin/complex V, reduction in protein carbonyls/MDA, increase in SOD2/GPx/catalase, and a shift towards lower levels of IL-1β/IL-6/TNF-α and higher levels of IL-11 in the hippocampus and prefrontal cortex. The work expands on previous literature on epicatechin by testing the (+) enantiomer in aged animals with parallel profiling of the hippocampus+PFC through behaviour, mitochondrial bioenergetics, redox status, and cytokines.

There are crucial major concerns that authors should address. Among these:

  1. Improvements in memory and oxidative/mitochondrial markers with epicatechin in aging/neuropathology models is currently well documented (from others and from the same authors, using the (−)-Epi from). Authors should then underline what is clearly new here, which cannot be simply the difference in the epitope. (i.e., in aged rats with parallel hippocampus + PFC profiling, including IL-11). Conclusions should be consistently tempered accordingly. Linked to the novelty, it would be helppul to add a focused comparison to prior (−)-Epi studies and the 2021 head-to-head (+) vs (−) mouse study. (PMID: 17537957; 33900336; 6115745). 
  2. Methods state “unpaired t-test followed by Tukey’s test,” which is internally inconsistent (Tukey is a post-hoc for ANOVA). It might be useful to include a multiple correlation analysis test, as multiple enzyme/biochemical measures were performed from the same mice (if the 6 mice reported are the same for all measurements). In this case, the analysis should use appropriate ANOVA factors or a clearly pre-specified primary endpoint with FDR control. Please re-analyze and report exact tests, factors, multiplicity strategy, and effect sizes (with CIs).

  3. Authors used n=6 mice per group. However, it is not indicated how this number of mice was determined. The number is quite small for behavioral + biochemical profiling across numerous outcomes. Authors must provide evidence about any usage of an a-priori power calculation for the behavioral primary endpoint (ORT) and discuss power for secondary biochemical measures; consider increasing n or tempering conclusions.
  4. Procedure for hippocampal region dissection is not reported in the material and methods. Was this a new metod or refers to previous works (method established?).
  5. Enzyme “activities” are reported as “nmol per mg protein” without time (e.g., catalase/SOD2/GPx). Standard reporting for activities includes rate units (e.g., nmol·min⁻¹·mg⁻¹ protein); citrate synthase is typically μmol or nmol·min⁻¹·mg⁻¹. Please clarify units, calculation formulas, and kit conversions, and ensure consistency with accepted practice (PMID: 12531911; 11208573).
  6. Only males were studied. Please justify and discuss generalizability per ARRIVE/NIH SABV expectations. Was female variability/estrous a factor for this sex selection?
  7. Authors stated that The scoring was made in a plinded manner. However, it is unclear whether tissue processing, ELISAs/Luminex, activity assays, ATP assays, and Western densitometry were performed and analyzed in a blind manner as well, as well as in a randomized order. Please detail blinding, plate layout/batch handling, and whether technical replicates were averaged before statistics.
  8. Authors framed IL-11 as anti-inflammatory. CNS studies show neuroprotection in stroke models, but a large body of work in other tissues shows pro-fibrotic/pro-inflammatory IL-11 signaling. (PMID: 31096144; 31554736; 33262481; 36012165). Please justify focusing on IL-11 (instead of canonical IL-10), discuss bidirectional evidence, and avoid over-interpreting IL-11 as purely anti-inflammatory. Authors should measure IL-10 or expanding limitations. 
  9. “Data available on request” is below current expectations. Please deposit raw trial-level ORT data, biochemical raw values, ATP luminescence files, and uncropped Western blots in a repository and update the Data Availability Statement (this is in accordance with MDPI Guidelines and reproducibility of the results).
  10. Figure 4F: it is stated that 6 samples were used, but there are only 4 Ctrl and 4 +Epi in the Western blot. Please clarify and provide graphs using dot + bar plot in all figures (this way to show results must be applied to all graphs).
  11. For ORT, specify object sets, counter-balancing, side-bias handling, a-priori exclusion criteria (e.g., minimal exploration), and whether DI was tested vs chance and between groups; report total exploration times to rule out sampling differences. Provide key ANY-maze parameters for OFT. (PMID: 28892027; 22160349; 30531711; 30003107)

Minor concerns:

  1. Figure 1D: it is absolutely not clear what symbols means. The image is not helpful in understanding the experimental procedure and must be at leas better described in the figure legend. Line threshold are not described in the figure legends and must be explained.
  2. Typos/terminology. Multiple typos (e.g., “malondihaldeide”, “absobance”) and minor grammar—please proofread.
  3. Figures/reporting. Add individual points to bar plots, exact p values, and effect sizes with CIs; include OFT trajectory examples and representative ORT frames.
  4. References should be updated with recent independent epicatechin/neuroprotection and human-cognition papers. PMID: 6115745 (consider recent reviews and 2024–2025 primary reports)
  5. Methods specifics. Clarify homogenization buffers vs kit specs (e.g., TBARS compatibility), tissue mass variance, and normalization (per mg protein vs per mg tissue). For SOD2 isolation via KCN, note residual SOD1/SOD3 contributions and validation.
  6. include calibration curves (R², range), intra/inter-assay CVs, and batch details, if possible.
  7. The symbol alpha for “TNF-α” is not visible (seems in bold?).

Round 2

Reviewer 1 Report

Comments and Suggestions for Authors

No further comments

Author Response

No comments provided by Reviewer

Reviewer 2 Report

Comments and Suggestions for Authors

Authors addressed all my concerns, which improved the clarity and quality of the data here presented. From my side, I do not have additional concerns and I would like to recommend the work for publication

Author Response

No comments provided by Reviewer